# Effects of copper nanoparticles synthesized from the entomopathogen *Metarhizium robertsii* against the dengue vector *Aedes albopictus* (Skuse, 1894)

**Perumal Vivekanandhan**[1,2], **Kannan Swathy**[2], **Pittarate Sarayut**[1,2], **Patcharin Krutmuang**[2]*

**1** Office of Research Administration, Chiang Mai University, Chiang Mai, Thailand, **2** Faculty of Agriculture, Department of Entomology and Plant Pathology, Chiang Mai University, Chiang Mai, Thailand

* patcharin.k@cmu.ac.th

**Data Availability Statement:** All relevant data are within the paper and its Supporting Information files.

## Abstract

*Aedes albopictus*, known as the Asian tiger mosquito, is a significant vector for dengue fever, chikungunya, zika virus, yellow fever. Current control methods rely on chemical insecticides, which face challenges such as resistance, environmental harm, and impact on non-target species *Eudrilus eugeniae* and *Artemia salina*. This study evaluates the toxic effects of biogenic copper nanoparticles (CuNPs) synthesized using *Metarhizium robertsii* intracellular extract obtained from our previous research. The CuNPs were tested against *A. albopictus* and non-target species at 24 and 48 hours post-treatment. Results demonstrated that entomopathogenic fungi-derived CuNPs exhibited potent mosquitocidal activity, resulting in 97.33% mortality in larvae, 93.33% in pupae, and 74.66% in adults at 48 hours post-treatment. The CuNPs derived from *M. robertsii* showed lower $LC_{50}$ values of 74.873 mg/L in larvae, 76.101 mg/L in pupae, and 136.645 mg/L in adults at 48 hours post-treatment. Additionally, 12 hours post-treatment, catalase (an antioxidant enzyme) activity decreased 1.5-fold in a dose-dependent manner, while glutathione S-transferase (a detoxification enzyme) activity increased 7.8-fold. CuNPs demonstrated lower toxicity to non-target species, with 24% mortality in *A. salina* and 24.44% mortality in *E. eugeniae* at 24 hours post-treatment. The $LC_{50}$ values were 634.747 mg/L for *A. salina* and 602.494 mg/L for *E. eugeniae* at 24 hours post-treatment. These findings indicate that entomopathogenic fungi-derived CuNPs are a promising, target-specific candidate for controlling *A. albopictus* at various life stages (larvae, pupae, and adults).

## Introduction

*Aedes albopictus*, commonly known as the Asian tiger mosquito, is a highly invasive species and a significant public health threat worldwide [1]. *A. albopictus* serves as a primary vector for several arboviruses, including dengue, Zika, chikungunya, and yellow fever, which pose

**Funding:** This research was supported by the Office of Research Administration, Chiang Mai University, Chiang Mai 50200, Thailand, under grant number EX010082.

**Competing interests:** The authors have declared that no competing interests exist

serious health risks globally [2–6]. In the current days the chemical control methods, such as insecticides and larvicides, are commonly used to manage mosquito populations [7,8]. These chemicals target different life stages of mosquitoes, aiming to reduce their numbers and disrupt disease transmission. However, these methods have significant drawbacks. Repeated use of insecticides has led to resistance development in mosquito populations, reducing their effectiveness [7,8]. Additionally, insecticides can adversely affect non-target species, including beneficial insects, birds, and aquatic organisms, disrupting local ecosystems and biodiversity [9–11]. Furthermore, chemical control can pose health risks to humans and animals through direct exposure or residual contamination in water and soil, leading to long-term ecological harm.

Nanotechnology offers innovative solutions for pest control [12–14]. Nano-insecticides use nanoparticles, typically ranging from 1 to 100 nm, which can be engineered for enhanced efficacy, stability, and specificity [12–17]. These nanoparticles deliver active ingredients more efficiently, penetrate biological barriers, and provide controlled release, thereby minimizing environmental impact. They can also target specific pest species, reducing harm to non-target organisms and addressing issues like resistance, offering a promising approach to sustainable pest management [12].

Entomopathogenic fungi-derived nano-insecticides provide a novel approach to mosquito control by combining biological and nanotechnological advantages [12,14]. Fungi such as Metarhizium and Beauveria fungal species naturally infect and kill insects [18–25]. When formulated into nanoparticles, fungal extracts can enhance stability, efficacy, and delivery. These nano-insecticides effectively target mosquito larvae and adults, reducing populations and disease transmission. They also offer an environmentally friendly alternative to chemical insecticides by minimizing resistance development and non-target effects [20–23]. This approach represents a promising, sustainable solution for managing mosquito-borne diseases. The nanoparticles enhance fungal efficacy by penetrating the mosquito cuticle and delivering fungal spores or bioactive compounds, leading to infection and disruption of physiological processes.

Enzymes such as catalase and glutathione S-transferase (GST) in mosquitoes play crucial roles in detoxification and defense against oxidative stress [26,27]. Catalase breaks down hydrogen peroxide into water and oxygen, while GST detoxifies xenobiotics by conjugating them with glutathione [28–30]. When mosquitoes are treated with nanoparticles, these enzymes are often upregulated as a defense mechanism against oxidative stress and potential toxicity. However, the efficacy of nanoparticle-based insecticides can overwhelm these defenses, leading to increased mortality. Understanding the roles of catalase and GST is essential for designing more effective nanoparticle formulations for mosquito control [31–33].

*Eudrilus eugeniae*, the African nightcrawler, is an important earthworm species for soil health and fertility due to its efficient decomposition of organic matter [34]. *Artemia salina*, or brine shrimp, is crucial in aquatic ecosystems and is widely used in aquaculture [35]. Both species are significant bioindicators for environmental studies. Evaluating the impact of nanoparticle-based insecticides on non-target species like *E. eugeniae* and *A. salina* is vital, as unintended harm to these organisms can disrupt soil and aquatic ecosystems. This study evaluates the toxic effects of nanoparticles, adapted from our previous research [12], on the larvae, pupae, and adults of *Aedes albopictus* after treatment with copper nanoparticles (CuNPs) derived from *Metarhizium robertsii*. The effects of these CuNPs on catalase and glutathione-S-transferase enzymes were also assessed, along with their impact on non-target species such as *Eisenia eugeniae* and *Artemia salina* (Fig 1).

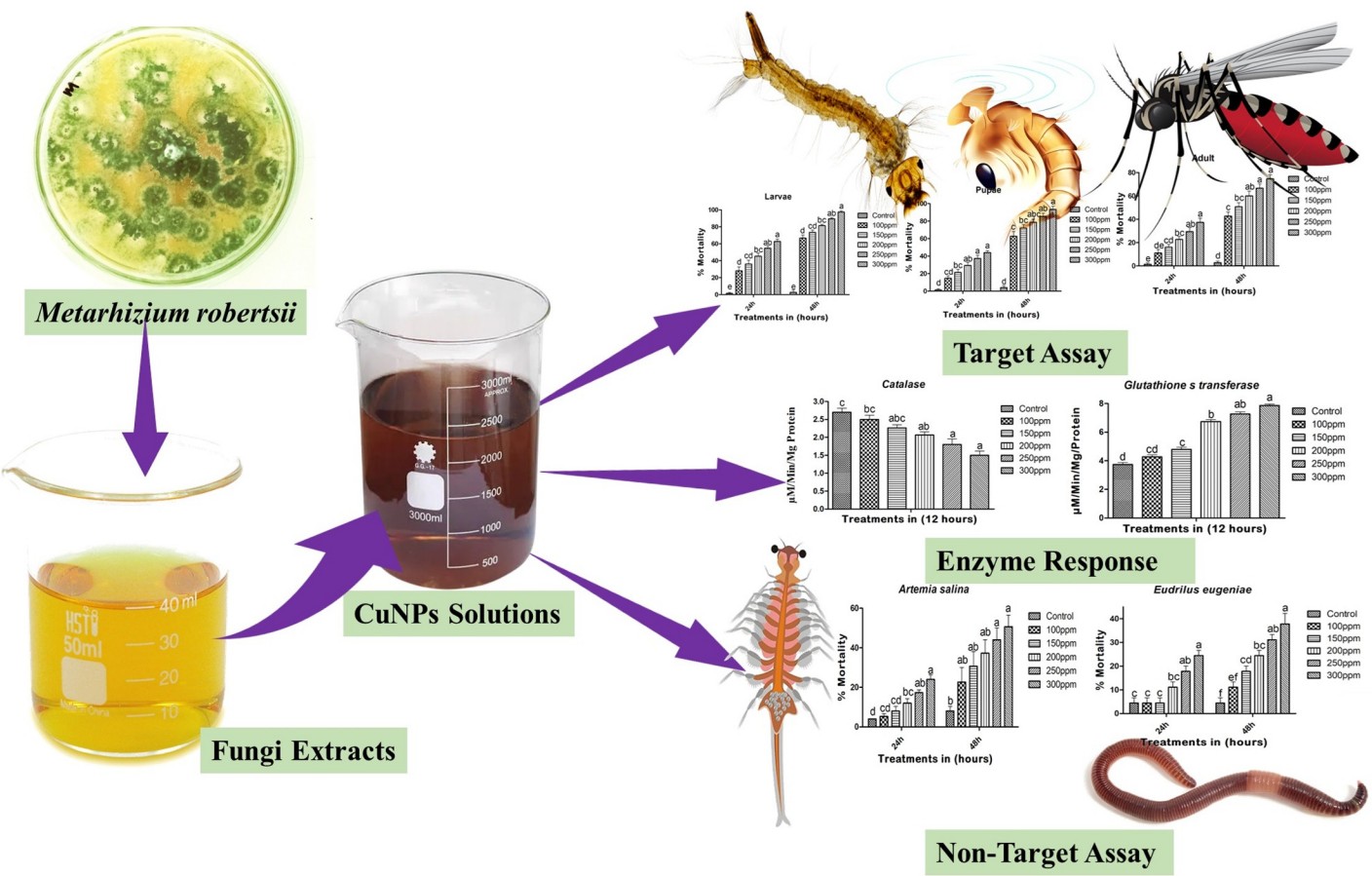

**Fig 1. Schematic diagram of entomopathogen derived copper nanoparticles (CuNPs) and their toxic effects on target and non-target species.**

## Materials and methods

### Fungal broth culture

The *M. robertsii* was isolated from the soil and broth was prepared following the method described in our previous studies [12]. Cultures were initially grown on potato dextrose agar (PDA) in 250 mL culture flasks containing 150 mL of potato dextrose broth (PDB). Chloramphenicol, at a concentration of 0.7 g/L, was added as an antibacterial agent. A spore suspension with a concentration of $1 \times 10^6$ spores/mL was then inoculated into the broth. After inoculation, the cultures were incubated at $28 \pm 2 °C$ for 12 days, as determined from our previous experiments [12].

### Fungal extract preparation and copper nanoparticles (CuNPs) synthesis

Fifty grams of *M. robertsii* fungal biomass was washed twice with sterile water, then placed in a 250 mL conical flask containing 150 mL of sterile distilled water. The flask was incubated for five days, followed by cold extraction on an orbital shaker at 170 rpm. After five days, the aqueous secondary metabolites were filtered using Whatman No. 1 filter paper (HiMedia, Maharashtra, India).

Fifteen millilitres of *M. robertsii* fungal culture filtrate was added to 85 mL of a 1 mM copper sulfate solution, mixed thoroughly, and heated at 60°C using a hot plate magnetic stirrer

(LabFriend India Private Limited, New Delhi, India) for 7 hours. After heating, the solution was incubated in the dark at $28 \pm 2°C$ for 72 hours, during which its color changed to dark brown. The solution was then filtered using Whatman No. 1 filter paper and washed multiple times with double-distilled water through centrifugation (Kesar Control Systems, Gujarat, India) at 11,000 rpm for 8 minutes to remove any unwanted particles. This washing process was repeated several times to ensure complete removal of impurities. Finally, the pellet was air-dried at room temperature for two days, after which the dried pellets were used for all subsequent experiments and catechization as detailed in our previous research [12].

## Mosquito culture and rearing

The larvae, pupae, and adults of *A. albopictus* mosquitoes were utilized in this study. These mosquitoes were successfully collected from the Chiang Mai University campus and maintained in the Insect Pathology Laboratory, Department of Entomology and Plant Pathology, Faculty of Agriculture, Chiang Mai University. The mosquito culture maintenance conditions were kept at room temperature ($26\pm2°C$) with 75–85% relative humidity and a 14:10 light-dark photoperiod. The mosquitoes were housed in plastic boxes filled with tap water. The larvae were fed a diet of biscuits and yeast powder in a 3:1 ratio, while adult mosquitoes were provided with 10% sugar solution containing vitamin B as food source to the adults.

## Non-target species

**Artemia salina.**   *A. salina* adults were cultured in a controlled laboratory environment using 1000 mL of seawater with a salinity of 30 parts per thousand. The photoperiod was controlled at 17:7 (light and dark) to maintain the pH between 8.0 and 8.5. A constant temperature of $27 \pm 1°C$ was maintained throughout the experiment. Aeration was provided every 24 hours using an aspirator to ensure optimal conditions for larval growth.

**Eudrilus eugeniae.**   Earthworms (*E. eugeniae*) were reared in plastic containers measuring 30 cm in length, 20 cm in width, and 20 cm in height. The containers were kept at a consistent temperature of $26 \pm 1°C$. Cow dung and goat manure were provided as offered sources, supplying the providing essential the earthworms. The containers were monitored regularly checked the uphold environment, ensuring that the earthworms had consistent access continuous food and suitable living appropriate for their development and growth well-being.

## Mosquitocidal bioassay

**Larval and pupicidal toxicity assay.**   The toxicity of microbial-synthesized copper nanoparticles (CuNPs) was assessed against the larvae and pupae of *A. albopictus* using a multi-concentration test. This test included a control and various concentrations (100, 150, 200, 250 and 300 mg/L) of CuNPs test solution. The larvicidal and pupicidal tests involved placing 25 4th instar larvae and 25 pupae individually in a multi-vial tray filled with 249 mL of sterilized double-distilled water and added 1 mL of the CuNPs solutions. Double-distilled water was used as a solvent to dilute the nanoparticle solutions to the appropriate concentrations (100, 150, 200, 250 and 300 mg/L). Each concentration was tested in three replicates, with each replicate containing 25 larvae and 25 pupae. Distilled water was used as the negative control. After 24 and 48 hours post-treatment, larval and pupal mortality were calculated using Abbott's formula [36].

**Adulticidal toxicity assay.**   The experiments involved adult *A. albopictus* mosquitoes (4–7 days old) that were fed a 10% sugar solution. Different concentrations (100, 150, 200, 250 and 300 mg/L) of copper nanoparticles (CuNPs) were prepared by diluting the CuNPs with double-distilled water. Filter papers (150×130 mm) were individually soaked in the diluted copper nanoparticle solutions. A control paper containing only double-distilled water was also

prepared. The papers were left to air-dry overnight at room temperature and then used as freshly soaked papers for the adulticidal bioassay.

The bioassay was conducted using a kit consisting of two cylindrical plastic tubes (130×50 mm each) following the methodology outlined by WHO [37]. One tube was utilized to expose the mosquitoes to the CuNPs, while the other tube was used to house the mosquitoes before and after exposure. The nanoparticle-impregnated papers were rolled and placed into the exposure tube, which was then sealed at one end with a 15-mesh wire screen. A total of 25 mosquitoes, fed sucrose and deprived of blood, were introduced into the exposure tube. The mortality effects of the CuNPs were observed at 12-hour and 48-hour intervals. Over the 48-hour incubation period, cotton boll soaked with a 10% sugar solution containing vitamin B complex were inserted into the tube. Mosquito mortality was documented after 24 and 48 hours. Both the control and treated groups (with CuNPs) were replicated three times for each concentration. Each replicate consisted of 25 insects.

**Enzyme response.** Following a 12-hour experiment on *A. albopictus* larvae, both the control group and the group treated with copper nanoparticles (CuNPs) had their live larvae rinsed with double-distilled water and dried with tissue paper to eliminate any remaining moisture. Afterwards, the larvae were crushed into a uniform mixture in 50 mL of sodium phosphate buffer (20 mM, pH 7.0) that had been cooled on ice. This was done to measure the activity of the enzymes. The homogenates were subsequently subjected to centrifugation at a speed of 13,000 revolutions per minute for a duration of 20 min at a temperature of 4°C. The supernatant obtained was collected for subsequent analysis. The standardized samples were kept at a temperature of 4°C until they were utilized in subsequent experiments.

**Catalase enzyme assay (CAT).** The catalase (CAT) enzyme assay followed the methods outlined by Wang et al. [38]. Solution-A (50 mM $KPO_4$, pH 7.0) and Solution-B (0.036% $H_2O_2$ in $KPO_4$) were prepared. Specifically, 2.9 mL of Solution-A and Solution-B were combined, comprising 0.03 mL of $H_2O_2$ mixed with phosphate buffer. Subsequently, 0.1 mL of larval midgut tissue homogenate was added to the solution, and this mixture was compared against a control cuvette containing only $H_2O_2$ and phosphate buffer. The catalase enzyme activity was determined by measuring the absorbance at 240 nm.

**Glutathione transferase enzyme assay (GST).** The experimental procedure consisted of filling a cuvette with 820 μL of phosphate buffer (66 mmol/L, pH 7.0), adding 35 μL of enzyme solution, 105 μL of 50 mmol/L glutathione (GSH), and 40 μL of 30 mmol/L 1-chloro-2,4-dinitrobenzene (CDNB) to evaluate enzymatic activity. Absorbance changes were measured at a wavelength of 340 nm for a duration of 5 minutes. The results were measured in terms of nanomoles of GSH-CDNB generated per minute per milligram of protein. The process was iterated three times for the purpose of validation.

## Non-target bioassay

**Toxicity on *A. salina*.** The toxicity of copper nanoparticles (CuNPs) was assessed on mature *A. salina* at concentrations ranging from 100, 150, 200, 250 and 300 mg/L. Twenty-five shrimp were transferred to 249 mL of seawater, to which 1 mL of different CuNPs test concentrations was individually added into respective test containers containing *A. salina*. Sterile seawater was used as the negative control. The experiment included three replicates for each concentration, and the mortality of *A. salina* was recorded at 24 and 48 hours post-treatment. Morphological changes in *A. salina* were observed under a binocular microscope at both 24 and 48 hours after treatment.

***Eudrilus eugeniae* filter paper bioassay.** The filter paper contact method, adapted from OECD Guideline No. 207, utilized 7 cm diameter Petri dishes. Different concentrations of

CuNPs (mg/L) were suspended in double-distilled water and applied to the filter paper surfaces. CuNP suspensions were freshly prepared for each experiment using vortex mixing (30 seconds, 1800 rpm) and sonication (30 min). Prior to application, each suspension was briefly vortexed for 5 seconds. Five milliliters of the suspension was pipetted onto the filter paper to moisten it. Adult earthworms (*E. eugeniae*, 250±30 mg) were used in the experiments. In the control group, earthworms were placed on filter paper treated with distilled water following the same preparation method. Fifteen earthworms were individually placed in each Petri dish, and three replicates were used for each treatment. After placing the earthworms, each dish was covered with cling film pierced with small ventilation holes. All Petri dishes were then placed in a large, plastic-covered glass container maintained at approximately 85% relative humidity and incubated in darkness at 25˚C. Mortality of the earthworms was recorded after 24 and 48 hours of exposure.

## Statistical analysis

The mortality rates of all stages of mosquitoes (larvae, pupae, and adults), *A. salina*, and *E. eugeniae* were determined 24 and 48 hours after treatment using Abbott's formula [36]. The experiments were replicated three times, and the results were reported as the average and standard deviation (SD). A one-way analysis of variance (ANOVA) was conducted, followed by Tukey's Honestly Significant Difference (HSD) test, to identify significant differences between the groups ($p < 0.05$). Probit analysis was conducted using SPSS-20 to estimate the $LC_{50}$ and $LC_{90}$ values, along with a 95% confidence interval.

## Results

### Larvicidal toxicity bioassay

Entomopathogenic fungi-derived copper nanoparticles (CuNPs) demonstrated significant mosquito larvicidal activity. At 24 hours post-treatment, the CuNPs showed a mortality rate of 62.66% ($F_{(5,12)} = 40.305$; $p < 0.01$), with $LC_{50}$ and $LC_{90}$ values of 216.726 (189.552–264.984) mg/L and 1016.916 (846.725–1256.845) mg/L, respectively. At 48 hours post-treatment, the CuNPs caused a 97.33% larval mortality rate ($F_{(5,12)} = 217.156$; $p < 0.01$), with $LC_{50}$ and $LC_{90}$ values of 74.873 (36.722–112.956) mg/L and 247.213 (210.531–284.003) mg/L, respectively (Fig 2).

### Catalase enzyme assay

Exposing *A. albopictus* larvae to copper nanoparticles (CuNPs) led to a 1.5-fold decrease in the levels of the larval catalase enzyme (see Fig 3). The catalase enzyme activity showed a substantial reduction at a concentration of 300 mg/L after 12 hours of treatment ($F_{(5,12)} = 9.763$; df = 5; $p \leq 0.001$). The reduction in catalase enzyme activity in the larvae exhibited a direct correlation with the concentration of copper nanoparticles. These findings indicate that the expression of enzymes is mainly influenced by the dosage. The control larvae did not show any alterations in enzyme levels, whereas the larvae exposed to higher concentrations of copper nanoparticles displayed a significant reduction in catalase enzyme levels (see Fig 3).

### Glutathione S-transferase enzyme assay

The results demonstrate a progressive rise in glutathione S-transferase (GST) activity when compared to the control treatments. Copper nanoparticles derived from entomopathogenic fungi effectively suppressed the activity of larval GST. The high concentrations that were tested, the larvae showed the greatest increase in GST levels after 12 hours of treatment at a

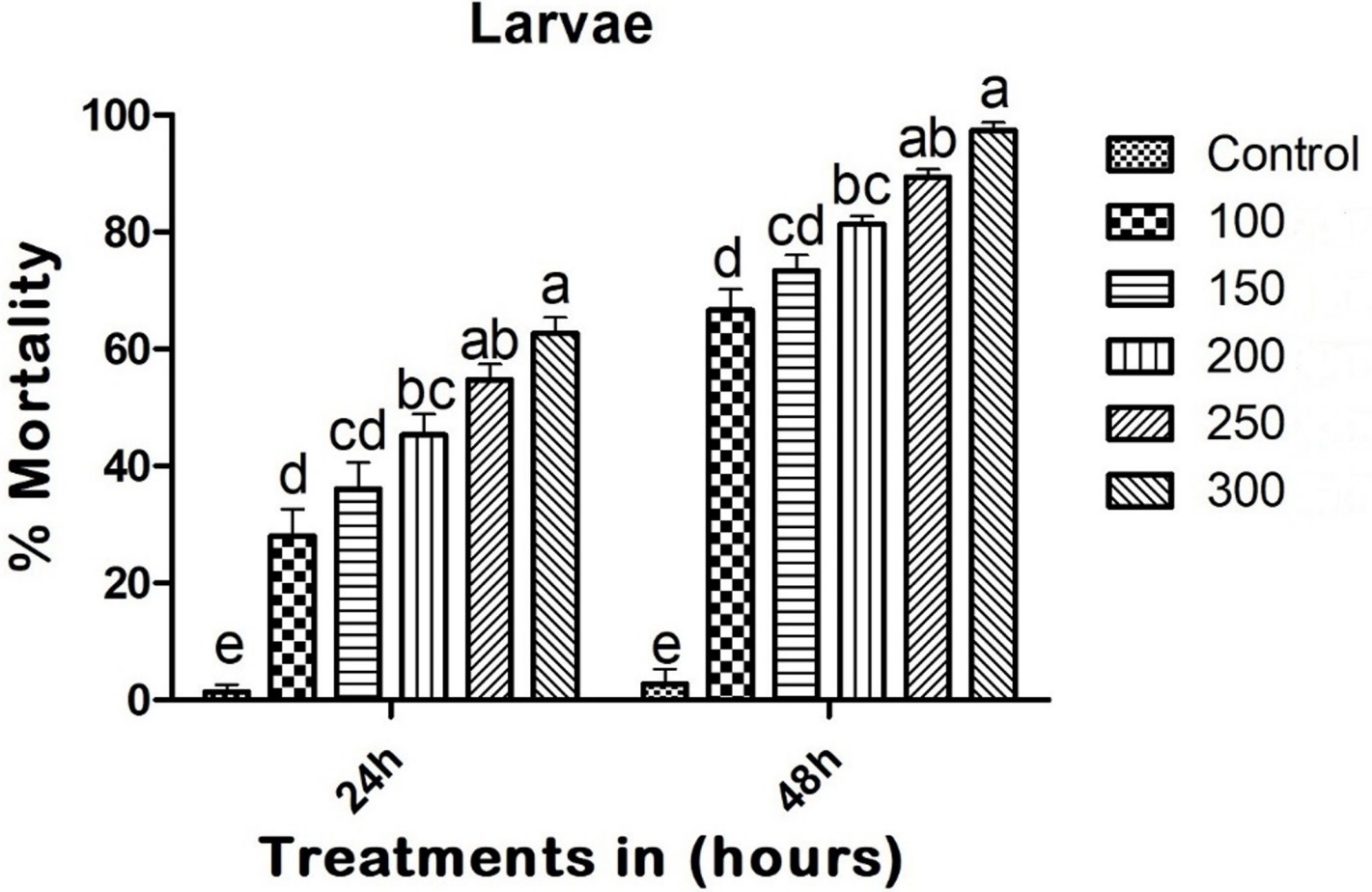

**Fig 2. Entomopathogenic fungi-derived copper nanoparticles (CuNPs) exhibited larvicidal activity against larvae of *A. albopictu* at 24 and 48h post treatment.** In a Tukey test, means (SE: Standard error) that are labeled with the same letters above the bars indicate no statistically significant difference (p ≤ 0.05). The test concentration unit was mg/L.

concentration of 300 mg/L ($F_{(5,12)}$ = 154.862; p ≤ 0.01) (see Fig 4). Furthermore, the increased concentration of 300 mg/L resulted in a gradual increase in GST activity by a factor of 7.86μm. The larvae exhibited an increased level of GST activity that corresponded to the highest concentration of copper nanoparticles. These findings indicate that the expression of the enzyme is primarily influenced by the dosage. Control larvae did not show any alterations in enzyme levels, whereas larvae treated with higher concentrations of copper nanoparticles exhibited a notable elevation in GST levels (see Fig 4).

### Pupicidal toxicity assay

Copper nanoparticles (CuNPs) demonstrated significant mosquito pupicidal activity. At 24 hours post-treatment, the CuNPs showed a mortality rate of 44% ($F_{(5,12)}$ = 29.037; p < 0.01), with $LC_{50}$ and $LC_{90}$ values of 369.932 (291.651–419.001) mg/L and 1710.702 (1428.110–2015.551) mg/L, respectively. At 48 hours post-treatment, the CuNPs caused a 93.33% pupal mortality rate ($F_{(5,12)}$ = 71.253; p < 0.01), with $LC_{50}$ and $LC_{90}$ values of 76.101 (53.750–88.531) mg/L and 299.985 (263.947–362.880) mg/L, respectively (Fig 5).

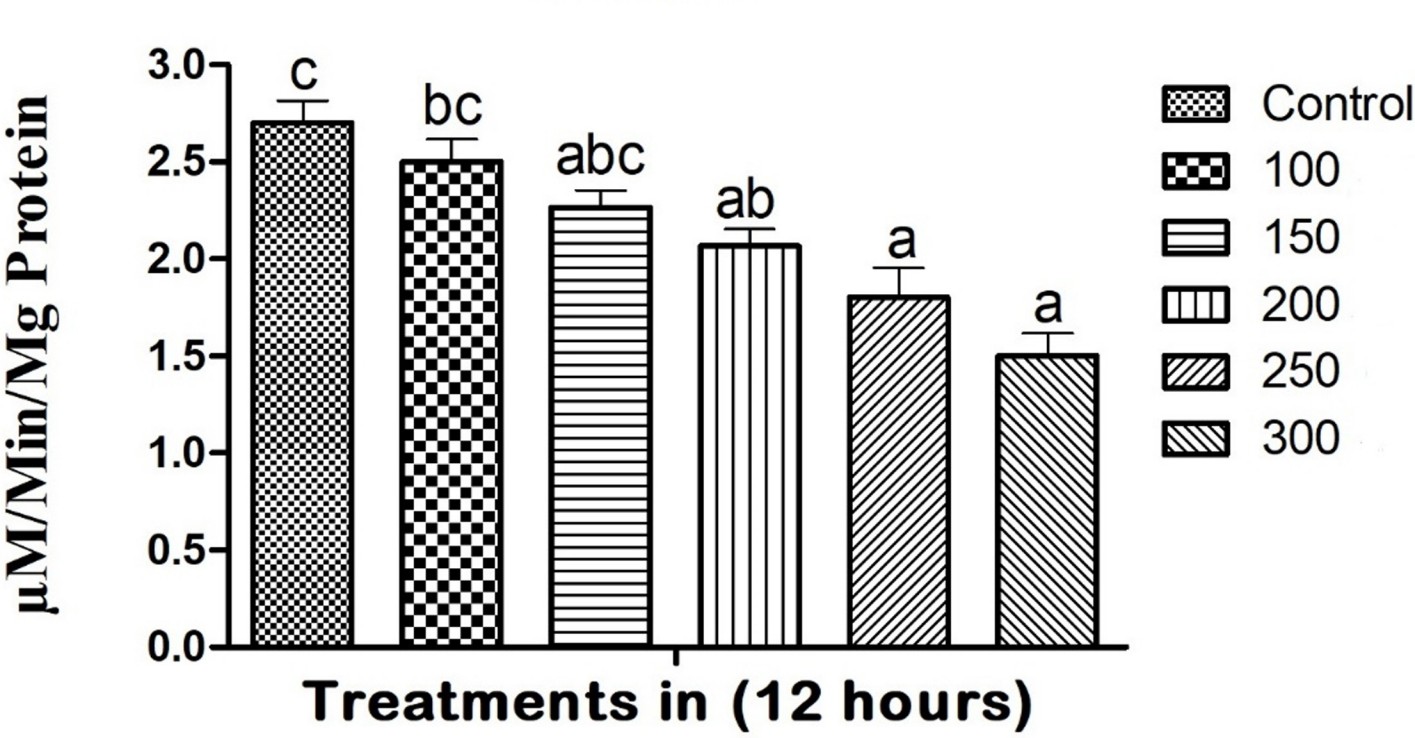

**Fig 3. The catalase enzyme response of *A. albopictus* larvae decreased (1.5-fold) in response to *M. robertsii*-derived copper nanoparticles.** In a Tukey test, means (SE: Standard error) that are labeled with the same letters above the bars indicate no statistically significant difference ($p \leq 0.05$). The test concentration unit was mg/L.

### Adult toxicity assay

Copper nanoparticles (CuNPs) demonstrated significant adulticidal activity against mosquitoes. Twenty-four hours after treatment, the CuNPs resulted in a mortality rate of 37.33% ($F_{(5,12)} = 25.539$; $p < 0.01$), with $LC_{50}$ and $LC_{90}$ values of 444.998 (329.005–482.932) mg/L and 1833.736 (1538.448–2036.883) mg/L, respectively. Forty-eight hours post-treatment, the CuNPs caused a 74.66% mortality rate in adults ($F_{(5,12)} = 58.258$; $p < 0.01$), with $LC_{50}$ and $LC_{90}$ values of 136.645 (123.740–166.372) mg/L and 746.212 (716.339–964.882) mg/L, respectively (Fig 6).

### Non-target effect on *A. salina*

Copper nanoparticles (CuNPs) demonstrated minimal toxicity effects on *A. salina*. Twenty-four hours after treatment, the CuNPs resulted in a mortality rate of 24% ($F_{(5,12)} = 26.400$; $p < 0.01$), with $LC_{50}$ and $LC_{90}$ values of 634.747 (594.277–683.004) mg/L and 2267.892 (2021.510–2369.410) mg/L, respectively. Forty-eight hours post-treatment, the CuNPs caused a 50.66% mortality rate in adults ($F_{(5,12)} = 6.155$; $p < 0.005$), with $LC_{50}$ and $LC_{90}$ values of 306.126 (292.642–347.990) mg/L and 1954.462 (1742.446–2116.812) mg/L, respectively (Fig 7).

### Non-target effect on *E. eugeniae*

Copper nanoparticles (CuNPs) demonstrated low toxicity effects on *E. eugeniae*. Twenty-four hours after treatment, the CuNPs resulted in a mortality rate of 24.44% ($F_{(5,12)} = 14.400$;

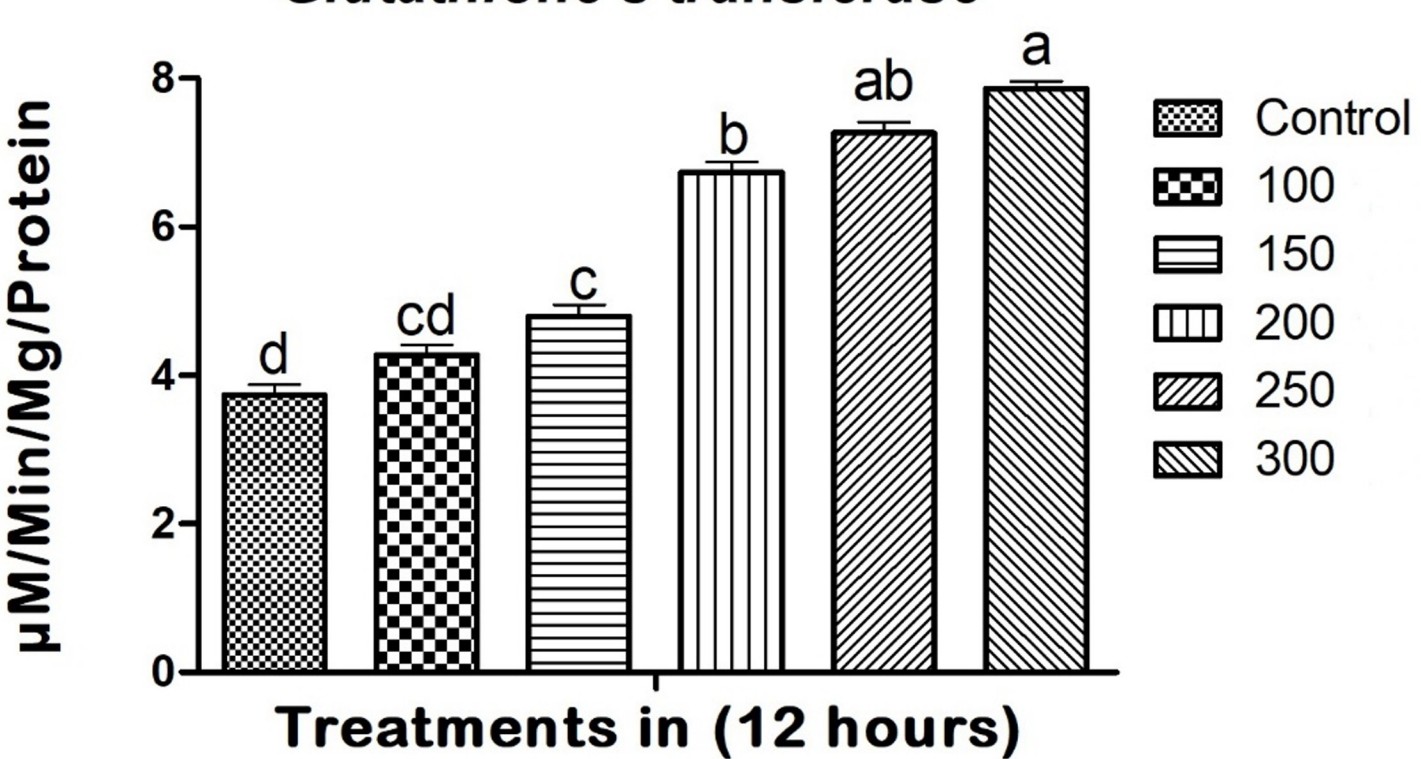

**Fig 4. The response of the *A. albopictus* larval Glutathione S-transferase (GST) enzyme increased 7.8-fold in response to *M. robertsii*-derived copper nanoparticles.** In a Tukey test, means (SE: Standard error) that are labeled with the same letters above the bars indicate no statistically significant difference (p ≤ 0.05). The test concentration unit was mg/L.

p < 0.01), with $LC_{50}$ and $LC_{90}$ values of 602.494 mg/L and 2022.188 mg/L, respectively. Forty-eight hours post-treatment, the CuNPs caused a 37.77% mortality rate ($F_{(5,12)}$ = 21.000; p < 0.005), with $LC_{50}$ and $LC_{90}$ values of 448.112 (415.836–478.152) mg/L and 2094.000 (2013.835–2264.704) mg/L, respectively (Fig 8). The positive control, monocrotophos at 100 mg/L, caused 55.55% and 97.77% mortality at 24 and 48 hours post-treatment.

## Discussion

The growing resistance of *A. albopictus* to conventional chemical insecticides highlights the need for alternative control strategies that are both effective and environmentally sustainable. This study explores the potential of biogenic copper nanoparticles (CuNPs), synthesized using the intracellular extract of *M. robertsii*, as a viable alternative. The discussion will focus on the implications of these findings and the mechanistic insights obtained from biochemical assays. The CuNPs synthesized in our previous research Vivekanandhan et al. [12], and this study demonstrate notable efficacy against *A. albopictus* at various life stages (larvae, pupae, and adults), with a strong impact on the larval stage, which is crucial for reducing the adult mosquito population and, consequently, the transmission of dengue and other vector-borne diseases (Figs 2–6). Additionally, similar research with silver nanoparticles derived from *Cochliobolus lunatus* fungi has shown high larvicidal potential against *A. aegypti* and *A. stephensi* [39].

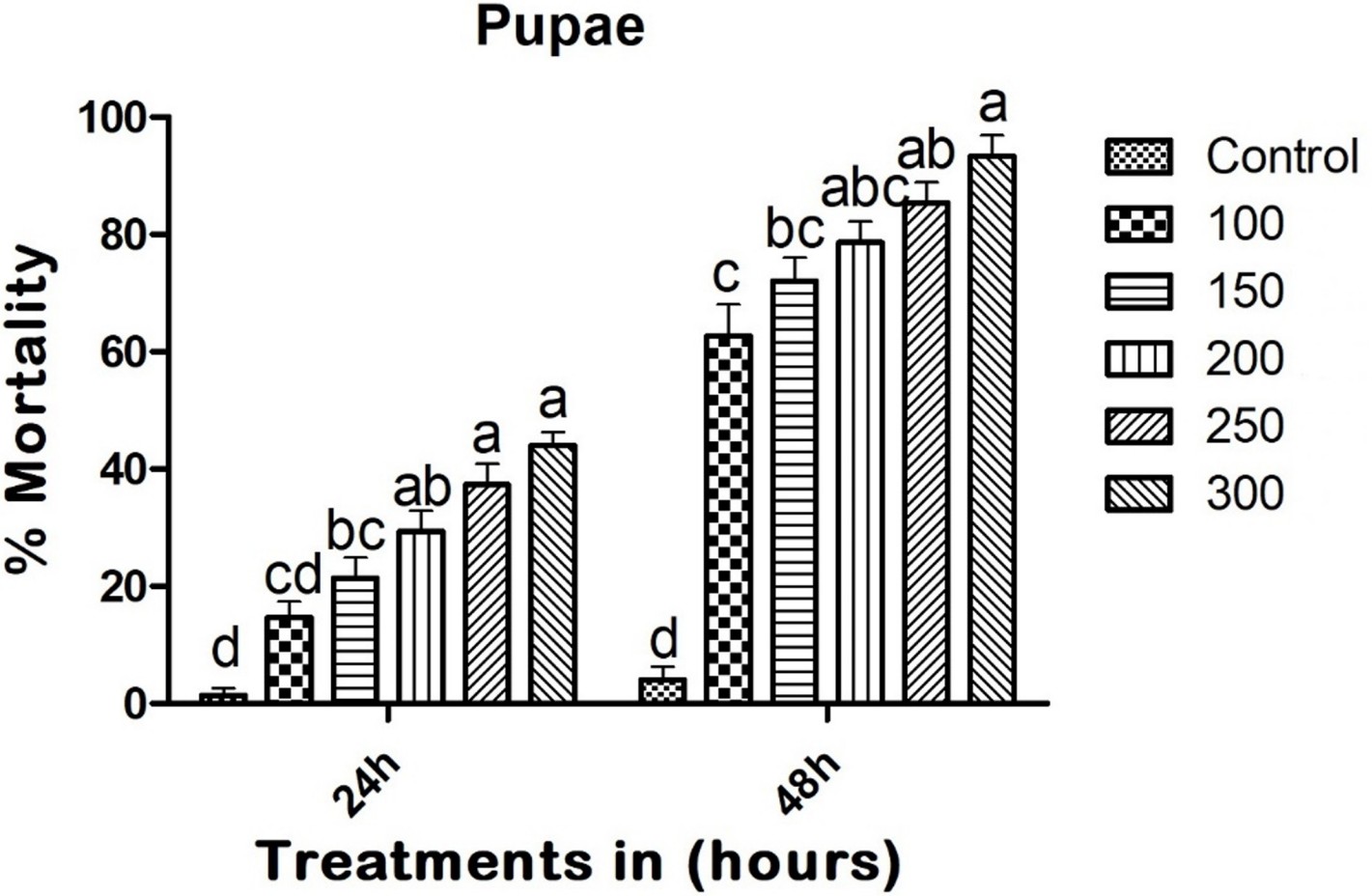

**Fig 5. Entomopathogenic fungi-derived copper nanoparticles (CuNPs) exhibited pupicidal activity against pupae of *A. albopictus* at 24 and 48h post treatment.** In a Tukey test, means (SE: Standard error) that are labeled with the same letters above the bars indicate no statistically significant difference (p ≤ 0.05). The test concentration unit was mg/L.

Soil-borne fungi *Aspergillus niger*-derived silver nanoparticles (AgNPs) exhibited promising larvicidal and pupicidal activities at minimal concentrations against larvae and pupae of *A. stephensi*, *C. quinquefasciatus*, and *A. aegypti* [40]. Green synthesis of silver nanoparticles using endolithic fungi *Talaromyces funiculosus* also demonstrated remarkable antimicrobial, anticancer, and mosquitocidal activities [41]. Similarly, entomopathogenic fungi *Fusarium oxysporum*-derived silver nanoparticles showed significant mosquito larvicidal activity against larvae of *Anopheles stephensi*, *Aedes aegypti*, and *Culex quinquefasciatus* [14].

Entomopathogenic fungi *Beauveria bassiana*-derived mycosynthesized silver nanoparticles exhibited notable mosquito larvicidal activity against the dengue vector, *Aedes aegypti* [42]. Likewise, *Metarhizium anisopliae*-derived zinc oxide and aluminum oxide nanoparticles demonstrated high mosquitocidal activity within a short time against *Culex quinquefasciatus* [43]. *Penicillium verucosum*-derived myco-synthesized silver nanoparticles (AgNPs) caused high mosquito larvicidal activity against larvae of the filarial mosquito vector, *Culex quinquefasciatus* [44]. *Chrysosporium tropicum* fungi-derived silver and gold nanoparticles showed significant mosquito larvicidal activities against larvae of *Culex quinquefasciatus* and *Anopheles stephensi* [45,46].

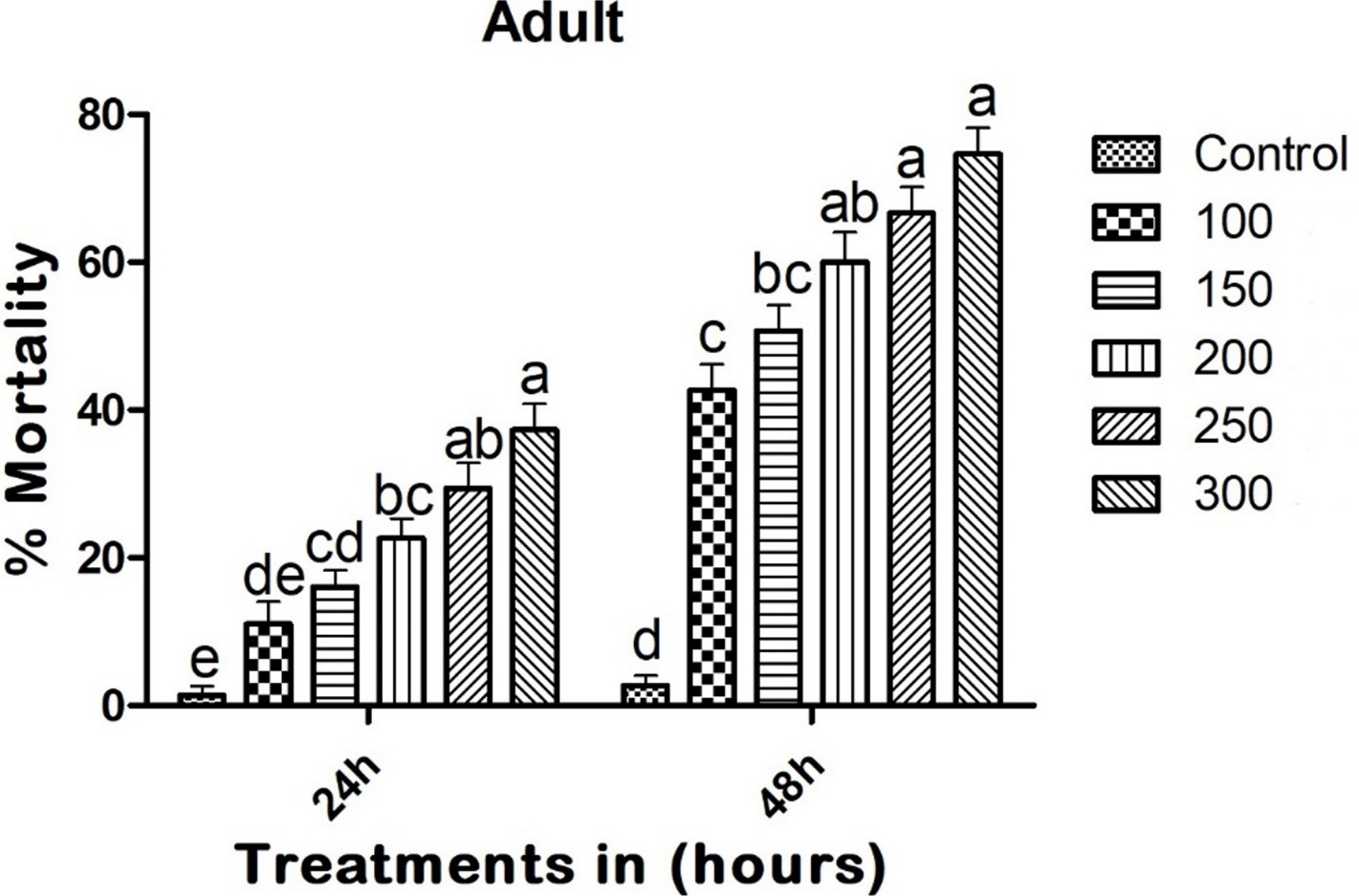

**Fig 6. Entomopathogenic fungi-derived copper nanoparticles (CuNPs) exhibited adulticidal activity against adult of *A. albopictus* at 24 and 48h post treatment.** In a Tukey test, means (SE: Standard error) that are labeled with the same letters above the bars indicate no statistically significant difference ($p \leq 0.05$). The test concentration unit was mg/L.

Mycosynthesized silver nanoparticles using *M. anisopliae* caused high mosquitocidal activity against the malaria vector *Anopheles culicifacies* [47]. *Penicillium chrysogenum*-derived magnesium oxide nanoparticles (Mg-NPs) showed remarkable insecticidal activity against larvae of *Anopheles stephensi* [48]. Similarly, soil fungi *C. tropicum*-derived silver and gold nanoparticles exhibited significant mosquito larvicidal activities against larvae of *Aedes aegypti*, with second instar larvae showing 100% mortality within 1 hour of treatment [45].

Mycosynthesized silver nanoparticles mediated by *Aspergillus niger* caused significant mosquito larvicidal activity against larvae of *Aedes aegypti* [49]. Entomopathogenic fungi *M. anisopliae*-derived silver nanoparticles demonstrated potential mosquitocidal activity against *Aedes aegypti* mosquitoes [50]. Similarly, *Metarhizium robertsii*-derived copper nanoparticles exhibited significant mosquito larvicidal activity against larvae of *A. stephensi*, *A. aegypti*, *C. quinquefasciatus*, and *T. molitor*, with minimal toxicity observed in non-target organisms such as *A. salina*, *A. nauplii*, *E. eugeniae*, and *E. andrei* [12]. The effectiveness of CuNPs can be attributed to their unique physicochemical properties, including small size and high surface area, which facilitate better interaction with biological membranes and cellular components. These properties likely enhance the penetration of CuNPs into the mosquito's body, leading to

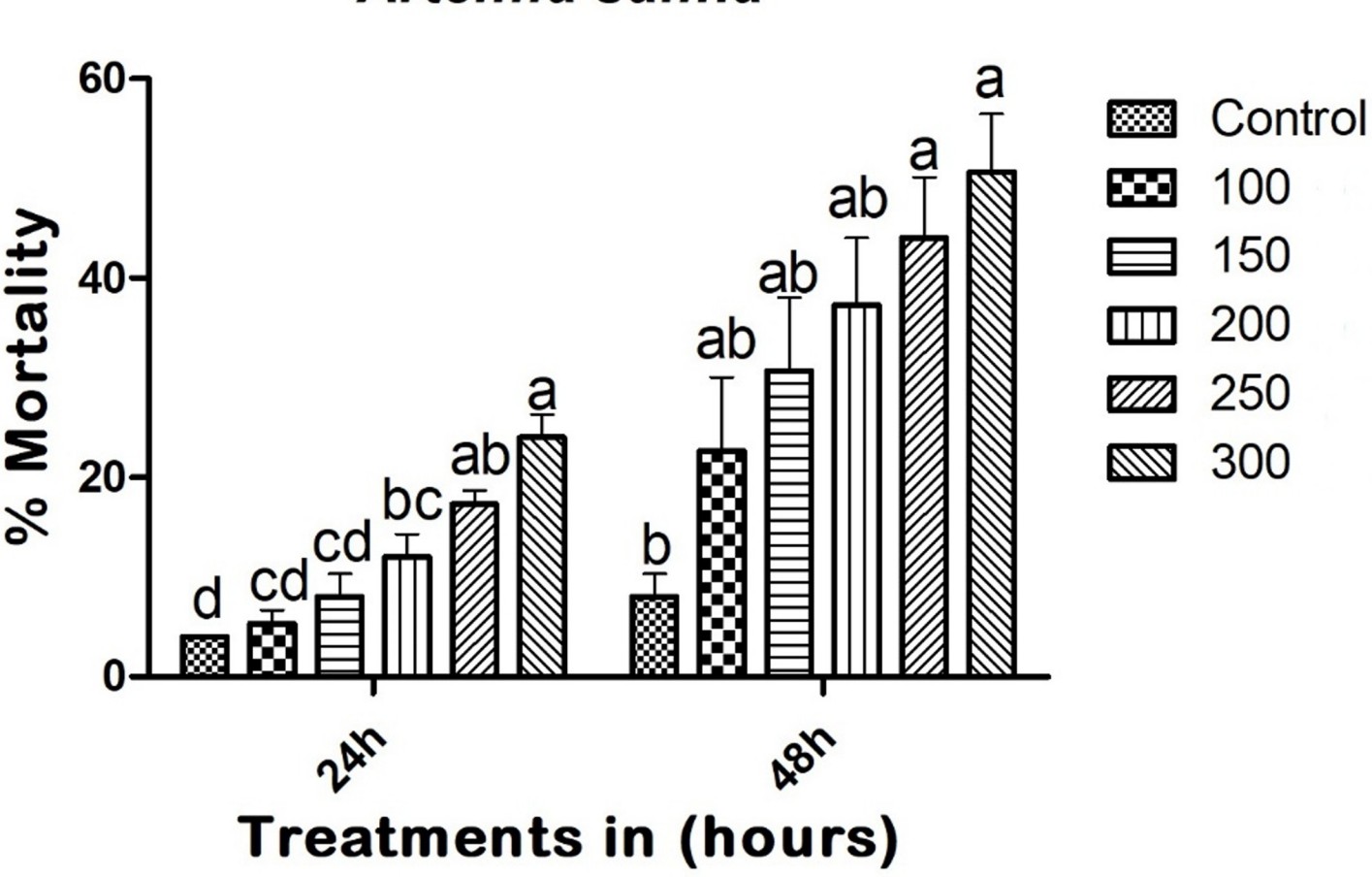

**Fig 7. Entomopathogenic fungi-derived copper nanoparticles (CuNPs) exhibited minimal toxicity effects against *A. salina* at 24 and 48h post treatment.** In a Tukey test, means (SE: Standard error) that are labeled with the same letters above the bars indicate no statistically significant difference (p ≤ 0.05). The test concentration unit was mg/L.

increased toxicity. Additionally, the involvement of the entomopathogenic fungus *M. robertsii* in the synthesis of CuNPs might confer additional bioactivity, potentially enhancing the nanoparticles' mosquitocidal properties.

Biochemical assays conducted in this study revealed significant alterations in enzyme activities post CuNPs treatment, providing insights into the potential mechanisms of toxicity. Catalase (CAT) activity decreased 1.5-fold in a dose-dependent manner 12 hours post-treatment (Fig 3), suggesting oxidative stress as a key factor in the CuNPs-induced toxicity. Catalase is crucial for mitigating oxidative damage by decomposing hydrogen peroxide into water and oxygen. A reduction in its activity implies an accumulation of reactive oxygen species (ROS), leading to oxidative stress and cellular damage. Similar to present study reported that entomopathogenic fungi *M. flavoviride* caused remarkable virulence against *S. litura* (Lepidoptera: Noctuidae) and its chemical constituents caused remarkable physiological and biochemical activities reported that entomopathogenic fungi *M. anisopliae* caused remarkable impact on antioxidant and detoxification enzymes of *S. frugiperda* [20–25].

Conversely, glutathione S-transferase (GST) activity increased 7.8-fold, indicating an upregulation of detoxification pathways in response to CuNPs exposure (Fig 4). GSTs play a vital

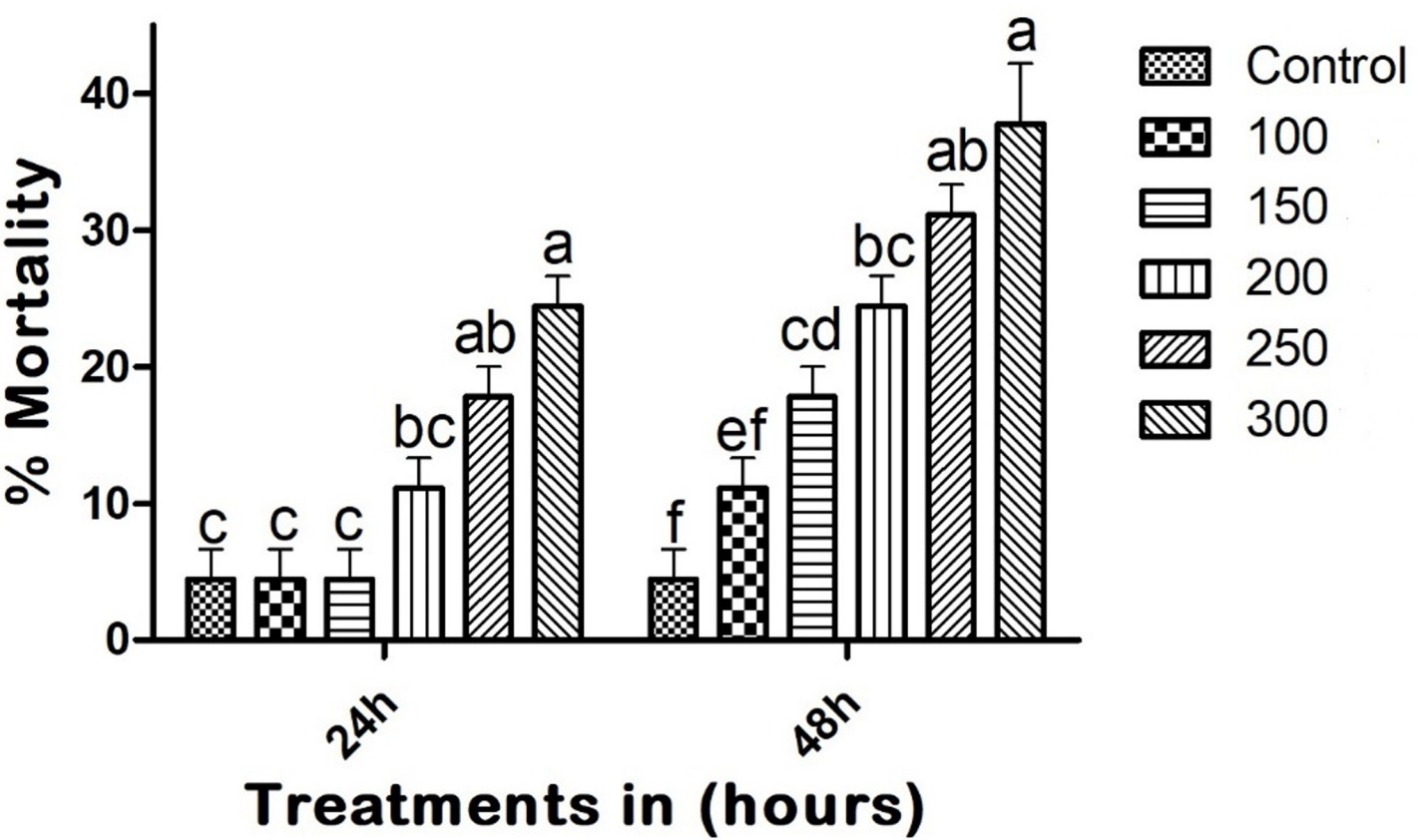

**Fig 8. Entomopathogenic fungi-derived copper nanoparticles (CuNPs) exhibited minimal toxicity effects against *E. eugeniae* at 24 and 48h post treatment.** In a Tukey test, means (SE: Standard error) that are labeled with the same letters above the bars indicate no statistically significant difference ($p \leq 0.05$). The test concentration unit was mg/L.

role in detoxifying endogenous and exogenous compounds by conjugating them with glutathione. The significant increase in GST activity suggests that *A. albopictus* attempts to counteract the toxic effects of CuNPs through enhanced detoxification processes. However, the overwhelming oxidative stress likely surpasses the detoxification capacity, leading to mortality. Similar to the present study, Vivekanandhan et al. [27] reported that the entomopathogenic fungus *M. flavoviride* exhibited remarkable virulence against *S. litura* (Lepidoptera: Noctuidae) and its chemical constituents caused significant physiological and biochemical changes. Additionally, Perumal et al. [24] demonstrated that *M. anisopliae* significantly impacted the antioxidant and detoxification enzymes of *S. frugiperda*. These findings highlight the dual role of oxidative stress and detoxification pathways in the toxic action of CuNPs. Targeting these pathways can enhance the efficacy of CuNPs and potentially lead to the development of synergistic control strategies that combine CuNPs with other agents that inhibit detoxification enzymes or amplify oxidative stress.

Entomopathogenic fungi, such as *Metarhizium* and *Beauveria* species, have significant effects on the detoxification enzymes of insects, including mosquitoes. Key enzymes like cytochrome P450 monooxygenases (P450s) and esterases, which are essential for the detoxification

of xenobiotics and insecticides, can be disrupted by fungal infections. Research suggests that these fungi interfere with detoxification pathways by inhibiting the activity of P450s and esterases, thereby reducing the insect's ability to metabolize and eliminate toxic substances. For example, exposure to fungal metabolites may downregulate P450 gene expression or impair enzyme function, increasing the mosquito's vulnerability to insecticidal compounds [51]. Additionally, esterases involved in insecticide resistance may also be inhibited, leading to greater susceptibility to fungal pathogens. This dual mechanism direct fungal pathogenicity and suppression of detoxification enzymes provides a promising approach for integrated mosquito control strategies [52].

One of the major advantages of biogenic CuNPs is their reduced toxicity to non-target species, as evidenced by the low mortality rates in *A. salina* (24%) and *E. eugeniae* (24.44%) at 24 hours post-treatment (Figs 7 and 8). The $LC_{50}$ values for *A. salina* (634.747 mg/L) and *E. eugeniae* (602.494 mg/L) were significantly higher than those for *A. albopictus*, indicating a wider safety margin for non-target organisms (Figs 7 and 8). This selectivity is crucial for minimizing ecological disruption and promoting the sustainability of pest control measures. Similar to the present study, Swathy et al. [18] demonstrated that the entomopathogenic fungus *B. bassiana* showed minimal effects on non-target species *E. eugeniae*. Likewise, Vivekanandhan et al. [21] reported that *M. rileyi* caused minimal effects on non-target species.

The biocompatibility of CuNPs can be attributed to their natural origin and the involvement of *M. robertsii* in their synthesis. Entomopathogenic fungi like *M. robertsii* are known for their specificity to insect hosts and minimal impact on non-target species. The use of such biological agents in nanoparticle synthesis can thus enhance the environmental safety profile of the resulting nanoparticles. Similarly, *M. robertsii*-derived copper nanoparticles showed no adverse effects on non-target species such as *A. salina*, *A. nauplii*, *E. eugeniae*, and *E. andrei*.

The promising results of this study position biogenic CuNPs as a valuable addition to integrated pest management (IPM) programs aimed at controlling *A. albopictus*. Their high efficacy, coupled with target specificity and environmental safety, make them suitable for use in conjunction with other control measures. For instance, CuNPs can be integrated with biological control agents, such as predators or pathogens of mosquitoes, to achieve synergistic effects. Additionally, their application can be timed to target the most vulnerable life stages of A. albopictus, enhancing overall control efficacy. Further research is necessary to optimize the formulation and delivery methods of CuNPs for field applications. Factors such as nanoparticle stability, persistence in the environment, and potential development of resistance should be thoroughly investigated. Moreover, large-scale field trials are essential to validate laboratory findings and assess the practical feasibility of using CuNPs in diverse ecological settings. While the findings of this study are promising, several limitations must be acknowledged. The laboratory conditions under which the experiments were conducted may not fully replicate field conditions, where environmental factors such as temperature, humidity, and presence of organic matter can influence the efficacy of CuNPs. Therefore, field trials are imperative to confirm the laboratory results and ensure the practical applicability of CuNPs in real-world scenarios.

## Conclusion

This study highlights the potential of biogenic copper nanoparticles (CuNPs) synthesized using *M. robertsii* intracellular extract as an effective and environmentally friendly alternative to chemical insecticides for controlling *A. albopictus*, a primary vector for dengue and other diseases. The CuNPs exhibited significant mosquitocidal activity, resulting in high mortality rates observed across various life stages of *A. albopictus*. The $LC_{50}$ values obtained indicate the effectiveness of these CuNPs at relatively low concentrations, especially in larvae and pupae.

Biochemical assays revealed that CuNPs influence antioxidant and detoxification enzyme activities in *A. albopictus*, suggesting a mechanistic basis for their toxicity. Importantly, the lower toxicity of these nanoparticles to non-target species such as *A. salina* and *E. eugeniae* underscores their potential as a target-specific control method, minimizing the ecological impact commonly associated with traditional chemical insecticides. Overall, entomopathogenic fungi-derived CuNPs offer a promising solution for integrated pest management programs aimed at reducing the prevalence of *A. albopictus* and mitigating the spread of vector-borne diseases. Further research and field trials are recommended to validate these findings and explore the practical applications of CuNPs in diverse environmental settings.

## Supporting information

**S1 Data.**
(XLSX)

## Acknowledgments

We would like to express our gratitude to the Department of Entomology and Plant Pathology for generously providing laboratory facilities, which greatly facilitated the progress and completion of this study.

## Author Contributions

**Conceptualization:** Perumal Vivekanandhan.

**Data curation:** Perumal Vivekanandhan, Kannan Swathy, Pittarate Sarayut, Patcharin Krutmuang.

**Formal analysis:** Perumal Vivekanandhan, Kannan Swathy, Pittarate Sarayut, Patcharin Krutmuang.

**Funding acquisition:** Perumal Vivekanandhan, Patcharin Krutmuang.

**Investigation:** Perumal Vivekanandhan.

**Methodology:** Perumal Vivekanandhan, Kannan Swathy.

**Project administration:** Perumal Vivekanandhan, Patcharin Krutmuang.

**Resources:** Perumal Vivekanandhan, Patcharin Krutmuang.

**Software:** Perumal Vivekanandhan, Kannan Swathy, Pittarate Sarayut.

**Supervision:** Perumal Vivekanandhan, Patcharin Krutmuang.

**Validation:** Perumal Vivekanandhan, Kannan Swathy, Patcharin Krutmuang.

**Visualization:** Perumal Vivekanandhan, Kannan Swathy.

**Writing – original draft:** Perumal Vivekanandhan, Kannan Swathy, Pittarate Sarayut.

**Writing – review & editing:** Perumal Vivekanandhan, Kannan Swathy, Pittarate Sarayut.

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
