## [Decision Letter · Decision Letter 0]

2 Sep 2024

PONE-D-24-33241Effects of Copper Nanoparticles Synthesized from Metarhizium robertsii against Dengue Vector Aedes albopictus (Skuse, 1894)PLOS ONE

Dear Dr. Krutmuang,

Thank you for submitting your manuscript to PLOS ONE. After careful consideration, we feel that it has merit but does not fully meet PLOS ONE’s publication criteria as it currently stands. Therefore, we invite you to submit a revised version of the manuscript that addresses the points raised during the review process.

**The manuscript needs minor revisions according the reviewers comments**

We look forward to receiving your revised manuscript.

Kind regards,

Shawky M Aboelhadid, PhD

Academic Editor

PLOS ONE

Journal Requirements:

2. Please include the full detail on nanoparticle synthesis to the Methods section, rather than referring to previous work.

Reviewers' comments:

Reviewer's Responses to Questions

**Comments to the Author**

1. Is the manuscript technically sound, and do the data support the conclusions?

Reviewer #1: Yes

Reviewer #2: Yes

2. Has the statistical analysis been performed appropriately and rigorously? 

Reviewer #1: Yes

Reviewer #2: Yes

3. Have the authors made all data underlying the findings in their manuscript fully available?

Reviewer #1: Yes

Reviewer #2: Yes

4. Is the manuscript presented in an intelligible fashion and written in standard English?

Reviewer #1: Yes

Reviewer #2: Yes

5. Review Comments to the Author

**Reviewer #1:** Dear authors,

This is an interesting research article dealing with the effects of copper nanoparticles synthesized from Metarhizium robertsii against Ae. albopictus larvae, pupae and adults and two non-target organisms. This study included also some biochemical assays about the effects of the tested nanomaterials on some enzymes.

I have the following comments for your consideration:

Lines 111-113: The scope of the manuscript should be more precise. It should be clarified that copper nanoparticles from the entomopathogenic fungus M. robertsii were used and that their effects on catalase and glutathione-s-transferase enzymes were evaluated as well.

Line 123: Please provide some info about blood feeding of mosquito females.

The laboratory conditions, i.e., temperature, humidity, and L:D, should be clarified for the laboratory bioassays against mosquito larvae, pupae and adults as well as non-target organisms.

In laboratory bioassays the metric units for ppm should be clarified, e.g., mg/L or μl/ml??

In laboratory toxicity bioassays, was it possible to include a control (blank) formulation without the fungus? Are you sure that the copper nanoparticles or other co-formulants had no toxic effect to mosquitoes?

Lines 172-174: Could you please clarify in the text why did you use pads soaked with vitamin B? Also, did you use untreated control for the adults? Please clarify in the text of the manuscript.

In the results section, a table with the results from probit analysis should be included for larvae, pupae and adults at 24h and 48h, presenting the LC50 values, confidential intervals, slope, and x2 values. In the footnote it should be clarified that LC50 values that overlap, differ significantly at α=0.05.

Is there any implication of entomopathogenic fungi on other detoxification enzymes of insects including mosquitoes such as P450s or esterases? If there is relevant information from the literature, this could be added in few lines to the discussion.

**Reviewer #2: **Comments to the Author

The manuscript titled “Effects of Copper Nanoparticles Synthesized from Metarhizium robertsii against Dengue Vector Aedes albopictus (Skuse, 1894)” by. Perumal et al. This study investigates the toxic effects of biogenic copper nanoparticles (CuNPs) synthesized using the intracellular extract of Metarhizium robertsii obtained from our previous research. The CuNPs were tested for their efficacy against the mosquito species Aedes albopictus and various non-target organisms. Toxicity assessments were conducted at 24 and 48 hours post-treatment to determine the potential of these biogenic CuNPs as a targeted and environmentally safe insecticidal agent. After carefully reviewing this manuscript I see merit in this research, this manuscript designed and well written with good grammar but it contains some errors so carefully revise entire manuscript with following comments;

1. Line 33-34: add some other disease name clearly.

2. Line 38: add non-target species name.

3. Line 41 and 42, 47: add test concentration unit ppm/ml or ppm/L

4. Line 92: add recent and relevant references.

5. Line 99: add a few more recent and relevant citations that will support your statement.

6. Line 109: add citations.

7. Line 117-119: rewrite the sentence.

8. Line 130: what biscuits? Provide the details.

9. Line 141: raised should be reared

10. Line 151 and 155: add all the test concentration clearly.

11. Line 214 and 215: what is 30 s and 5 s?

12. In the discussion don’t write your results in the discussion part, that is not the right place to write your results, also reputation from results part so carefully correct the errors in revised manuscript.

13. Reference section contains some typographical and formatting errors so carefully fix it.

14. Entire manuscript contains some grammatical errors so carefully correct errors in revised manuscript.

6. PLOS authors have the option to publish the peer review history of their article (what does this mean?). If published, this will include your full peer review and any attached files.

Reviewer #1: No

Reviewer #2: No

---

## [Author Response · Author response to Decision Letter 0]

10 Sep 2024

Journal Requirements:

2. Please include the full detail on nanoparticle synthesis to the Methods section, rather than referring to previous work.

• As per the journal requirement for the nanoparticle synthesis the our previous reference has been mentioned in the methodological part. Thank you.

• As per the journal requirement the errors has been fixed in the revised manuscript. Thank you.

• As per the journal requirement the errors has been fixed in the revised manuscript. Thank you.

• Dear professor, the research data will be sent on request to corresponding author so the error statement will be correct during revised manuscript submission. Thank you

• As per the journal requitement the corresponding author ORCID Id has been updated in the profile. Thank you.

• As per the journal requirement the reference section has been modified as per the based on the journal guideline. Thank you

Reviewer #1: 

 This is an interesting research article dealing with the effects of copper nanoparticles synthesized from Metarhizium robertsii against Ae. albopictus larvae, pupae and adults and two non-target organisms. This study included also some biochemical assays about the effects of the tested nanomaterials on some enzymes.

I have the following comments for your consideration:

• Thank you, professor, for your valuable comments as per your comments and suggestion the entire manuscript has been corrected in the revised manuscript. Thank you.

Lines 111-113: The scope of the manuscript should be more precise. It should be clarified that copper nanoparticles from the entomopathogenic fungus M. robertsii were used and that their effects on catalase and glutathione-s-transferase enzymes were evaluated as well.

• As per the reviewer comments the missing information has been added in the revised manuscript. Thank you.

Line 123: Please provide some info about blood feeding of mosquito females.

The laboratory conditions, i.e., temperature, humidity, and L:D, should be clarified for the laboratory bioassays against mosquito larvae, pupae and adults as well as non-target organisms.

• Dear professor, in this study we not provided ant blood to the mosquito adults we have provided 10% sugar solution to the adults that is sufficient for survival of mosquitoes and other missing information’s has been added in the revised manuscript. Thank you.

In laboratory bioassays the metric units for ppm should be clarified, e.g., mg/L or μl/ml??

• Dear professor in this study we used ppm/mL this information’s has been added in the revised manuscript. Thank you

In laboratory toxicity bioassays, was it possible to include a control (blank) formulation without the fungus? Are you sure that the copper nanoparticles or other co-formulants had no toxic effect to mosquitoes?

• Dear professor, in this study we not used any positive control, we have used only negative control, in the negative control also has some natural mortality not negative control effects.

Lines 172-174: Could you please clarify in the text why did you use pads soaked with vitamin B? Also, did you use untreated control for the adults? Please clarify in the text of the manuscript.

• Dear professor the 10% sugar solution containing vitamin B provided as a food source this this not any treatment, we provide this food source to both control and treatment groups. As per your comments this information’s has been provided in the revised manuscript. Thank you.

In the results section, a table with the results from probit analysis should be included for larvae, pupae and adults at 24h and 48h, presenting the LC50 values, confidential intervals, slope, and x2 values. In the footnote it should be clarified that LC50 values that overlap, differ significantly at α=0.05.

• Currently, we have presented the LC50 and LC90 p-values along with other relevant data. This manuscript contains 8 figures, and based on the reviewers' recommendations, some suggested not including tables. Therefore, we have omitted them from the manuscript. I hope you understand our situation.

Is there any implication of entomopathogenic fungi on other detoxification enzymes of insects including mosquitoes such as P450s or esterases? If there is relevant information from the literature, this could be added in few lines to the discussion.

• Thank you, professor, for your valuable suggestion, as per your suggestion we have added in the revised manuscript.

Reviewer #2: 

The manuscript titled “Effects of Copper Nanoparticles Synthesized from Metarhizium robertsii against Dengue Vector Aedes albopictus (Skuse, 1894)” by. Perumal et al. This study investigates the toxic effects of biogenic copper nanoparticles (CuNPs) synthesized using the intracellular extract of Metarhizium robertsii obtained from our previous research. The CuNPs were tested for their efficacy against the mosquito species Aedes albopictus and various non-target organisms. Toxicity assessments were conducted at 24 and 48 hours post-treatment to determine the potential of these biogenic CuNPs as a targeted and environmentally safe insecticidal agent. After carefully reviewing this manuscript I see merit in this research, this manuscript designed and well written with good grammar but it contains some errors so carefully revise entire manuscript with following comments;

• Thank you, dear professor, for your valuable suggestion, as per your suggestion the entire manuscript has been corrected in the revised manuscript. 

1. Line 33-34: add some other disease name clearly.

• As per the reviewer comments the missing information’s has been added in the revised manuscript.

2. Line 38: add non-target species name.

• As per the reviewer comments the missing information’s has been added in the revised manuscript. Thank you

3. Line 41 and 42, 47: add test concentration unit ppm/ml or ppm/L

• As per the reviewer comments the missing information’s has been added in the revised manuscript. Thank you

4. Line 92: add recent and relevant references.

• As per the reviewer comments the recent and relevant reference has been added in the revised manuscript.

5. Line 99: add a few more recent and relevant citations that will support your statement.

• As per the reviewer comments the recent and relevant reference has been added in the revised manuscript.

6. Line 109: add citations./

• As per the reviewer comments the recent and relevant reference has been added in the revised manuscript.

7. Line 117-119: rewrite the sentence.

• As per the reviewer comments the typographical errors has been corrected in the revised manuscript. Thank you

8. Line 130: what biscuits? Provide the details.

9. Line 141: raised should be reared

• As per the reviewer comments, the errors have been corrected in the revised manuscript. Thank you.

10. Line 151 and 155: add all the test concentration clearly.

• As per the reviewer comments the typographical errors has been corrected in the revised manuscript. Thank you.

11. Line 214 and 215: what is 30 s and 5 s?

• Dear professor s is seconds, the typographical errors has been corrected in the revised manuscript. Thank you.

12. In the discussion don’t write your results in the discussion part, that is not the right place to write your results, also reputation from results part so carefully correct the errors in revised manuscript.

• As per the reviewer comments the errors has been corrected in the reviewer comments. Thank you.

13. Reference section contains some typographical and formatting errors so carefully fix it.

• As per the reviewer comments the formatting and typographical errors has been corrected in the revised manuscript. Thank you

14. Entire manuscript contains some grammatical errors so carefully correct errors in revised manuscript.

• Thank you professor, as per your comments the entire manuscript the grammatical errors has been corrected by English expert.

---

## [Decision Letter · Decision Letter 1]

4 Oct 2024

PONE-D-24-33241R1Effects of Copper Nanoparticles Synthesized from the Entomopathogen Metarhizium robertsii Against the Dengue Vector Aedes albopictus (Skuse, 1894)PLOS ONE

Dear Dr. Krutmuang,

Thank you for submitting your manuscript to PLOS ONE. After careful consideration, we feel that it has merit but does not fully meet PLOS ONE’s publication criteria as it currently stands. Therefore, we invite you to submit a revised version of the manuscript that addresses the points raised during the review process.

**ACADEMIC EDITOR:****The authors need to correct and fix the units in the whole manuscrip**t.

We look forward to receiving your revised manuscript.

Kind regards,

Shawky M Aboelhadid, PhD

Academic Editor

PLOS ONE

**Journal Requirements:**

Reviewers' comments:

Reviewer's Responses to Questions

**Comments to the Author**

1. If the authors have adequately addressed your comments raised in a previous round of review and you feel that this manuscript is now acceptable for publication, you may indicate that here to bypass the “Comments to the Author” section, enter your conflict of interest statement in the “Confidential to Editor” section, and submit your "Accept" recommendation.

Reviewer #1: (No Response)

Reviewer #2: All comments have been addressed

2. Is the manuscript technically sound, and do the data support the conclusions?

Reviewer #1: Yes

Reviewer #2: Yes

3. Has the statistical analysis been performed appropriately and rigorously? 

Reviewer #1: Yes

Reviewer #2: Yes

4. Have the authors made all data underlying the findings in their manuscript fully available?

Reviewer #1: Yes

Reviewer #2: Yes

5. Is the manuscript presented in an intelligible fashion and written in standard English?

Reviewer #1: Yes

Reviewer #2: Yes

6. Review Comments to the Author

**Reviewer #1:** Dear authors,

Thank you for considering my comments. However, the following issues remain open.

What does ppm/ml mean? Normally, the concentrations should be expressed in ppm of CuNPs test solution to water solution and the metric units of ppm should be clarified, e.g., mg/lt or μl/lt. Please check.

You did not reply to my answer on how you are sure that copper does not contribute to the effect of copper nanoparticles of fungi to mosquitoes.

In the text it should be clarified that for the adulticidal toxicity an untreated control was also used and specify the replicates used.

Your response about the table with LC values is noted. However, I suggest adding in the text the confidential limits in parenthesis next to each LC50 or LC90 value.

**Reviewer #2: **Comments to the authors

The authors have responded to all the reviewer comments, and the manuscript now meets scientific standards. This revised manuscript is well-written and presents a novel study that contributes significantly to the field. The methodology is sound, and the data are clearly presented with appropriate analysis. The findings are both original and impactful, and the discussion effectively integrates the results within the broader context of existing literature. I now recommend this revised manuscript for publication.

7. PLOS authors have the option to publish the peer review history of their article (what does this mean?). If published, this will include your full peer review and any attached files.

Reviewer #1: No

Reviewer #2: No

---

## [Author Response · Author response to Decision Letter 1]

6 Oct 2024

Journal Requirements:

• Thank you, professor, for your valuable information, as per your information the retracted reference has been removed and new reference has been replaced in the revised manuscript. Thank you.

Reviewer #1: 

Thank you for considering my comments. However, the following issues remain open.

What does ppm/ml mean? Normally, the concentrations should be expressed in ppm of CuNPs test solution to water solution and the metric units of ppm should be clarified, e.g., mg/lt or μl/lt. Please check.

• Thank you, professor, for your comments, as per your comments we corrected the errors in the revised manuscript. Thank you.

You did not reply to my answer on how you are sure that copper does not contribute to the effect of copper nanoparticles of fungi to mosquitoes.

• Dear professor, I did not specifically claim that copper does not contribute to the effect. However, the unique properties of nanoparticles, such as surface area and reactivity, significantly differ from bulk copper's effects on mosquitoes.

In the text it should be clarified that for the adulticidal toxicity an untreated control was also used and specify the replicates used.

• Thank you, professor, as per your comments, the missing replicate information’s has been added in the revised manuscript. 

Your response about the table with LC values is noted. However, I suggest adding in the text the confidential limits in parenthesis next to each LC50 or LC90 value.

• Thank you, dear professor, as per your comment we have added the upper and lower confidential limits has been added in the revised manuscript. Thank you

Reviewer #2: 

The authors have responded to all the reviewer comments, and the manuscript now meets scientific standards. This revised manuscript is well-written and presents a novel study that contributes significantly to the field. The methodology is sound, and the data are clearly presented with appropriate analysis. The findings are both original and impactful, and the discussion effectively integrates the results within the broader context of existing literature. I now recommend this revised manuscript for publication.

• Thank you, Professor, for reviewing our research manuscript and for your valuable comments and suggestions. Your feedback has been incredibly helpful in improving the quality of our manuscript. We truly appreciate your kind support.

---

## [Decision Letter · Decision Letter 2]

16 Oct 2024

PONE-D-24-33241R2Effects of Copper Nanoparticles Synthesized from the Entomopathogen Metarhizium robertsii Against the Dengue Vector Aedes albopictus (Skuse, 1894)PLOS ONE

Dear Dr. Krutmuang, 

Thank you for submitting your manuscript to PLOS ONE. After careful consideration, we feel that it has merit but does not fully meet PLOS ONE’s publication criteria as it currently stands. Therefore, we invite you to submit a revised version of the manuscript that addresses the points raised during the review process.

We look forward to receiving your revised manuscript.

Kind regards,

Shawky M Aboelhadid, PhD

Academic Editor

PLOS ONE

Journal Requirements:

**Additional Editor Comments:**

The authors need to revise and fix the used units in the manuscript. 

Reviewers' comments:

Reviewer's Responses to Questions

**Comments to the Author**

1. If the authors have adequately addressed your comments raised in a previous round of review and you feel that this manuscript is now acceptable for publication, you may indicate that here to bypass the “Comments to the Author” section, enter your conflict of interest statement in the “Confidential to Editor” section, and submit your "Accept" recommendation.

Reviewer #1: (No Response)

2. Is the manuscript technically sound, and do the data support the conclusions?

Reviewer #1: Yes

3. Has the statistical analysis been performed appropriately and rigorously? 

Reviewer #1: Yes

4. Have the authors made all data underlying the findings in their manuscript fully available?

Reviewer #1: Yes

5. Is the manuscript presented in an intelligible fashion and written in standard English?

Reviewer #1: Yes

6. Review Comments to the Author

Reviewer #1: Dear authors,

Thank you for your reply. One last point is still open. Please indicate in the materials & methods of the manuscript the metric units of the tested material (e.g., in mg/L or μl/L) that are expressed in ppm.

7. PLOS authors have the option to publish the peer review history of their article (what does this mean?). If published, this will include your full peer review and any attached files.

Reviewer #1: No

---

## [Author Response · Author response to Decision Letter 2]

21 Oct 2024

Journal Requirements:

• Dear Sir/Madam,

In response to your comments, we have thoroughly checked the references and can confirm that no retracted articles have been cited. However, if you identify any references that have been retracted, please let us know, and we will promptly remove and replace them with recent and relevant citations.

Thank you for your assistance.

Additional Editor Comments:

The authors need to revise and fix the used units in the manuscript.

• Thank you, Professor, for your valuable comments. In line with the reviewer’s feedback, the errors have been corrected in the revised manuscript. I appreciate your guidance and support. 

Reviewer #1

Thank you for your reply. One last point is still open. Please indicate in the materials & methods of the manuscript the metric units of the tested material (e.g., in mg/L or μl/L) that are expressed in ppm.

• Thank you, Professor, for your valuable comments. Your feedback has been extremely helpful in strengthening our manuscript. As per your suggestions, the test concentration units have been added in the revised version. Once again, thank you for your support.

---

## [Editor Report · Decision Letter 3]

8 Nov 2024

Effects of Copper Nanoparticles Synthesized from the Entomopathogen Metarhizium robertsii Against the Dengue Vector Aedes albopictus (Skuse, 1894)

PONE-D-24-33241R3

Dear Dr. Krutmuang,

We’re pleased to inform you that your manuscript has been judged scientifically suitable for publication and will be formally accepted for publication once it meets all outstanding technical requirements.

Kind regards,

Shawky M Aboelhadid, PhD

Academic Editor

PLOS ONE
---

## [Editor Report · Acceptance letter]

15 Nov 2024

PONE-D-24-33241R3 

PLOS ONE

Dear Dr. Krutmuang, 

I'm pleased to inform you that your manuscript has been deemed suitable for publication in PLOS ONE. Congratulations! Your manuscript is now being handed over to our production team.

Kind regards, 

on behalf of

Professor Shawky M Aboelhadid 

Academic Editor

PLOS ONE